# Essential Oils and Their Compounds as Potential Anti-Influenza Agents

**DOI:** 10.3390/molecules27227797

**Published:** 2022-11-12

**Authors:** Ayodeji Oluwabunmi Oriola, Adebola Omowunmi Oyedeji

**Affiliations:** Department of Chemical and Physical Sciences, Faculty of Natural Sciences, Walter Sisulu University, Nelson Mandela Drive, P/Bag X1, Mthatha 5117, South Africa

**Keywords:** essential oils, flu-related diseases, influenza viruses, antiviral activity, anti-influenza agents

## Abstract

Essential oils (EOs) are chemical substances, mostly produced by aromatic plants in response to stress, that have a history of medicinal use for many diseases. In the last few decades, EOs have continued to gain more attention because of their proven therapeutic applications against the flu and other infectious diseases. Influenza (flu) is an infectious zoonotic disease that affects the lungs and their associated organs. It is a public health problem with a huge health burden, causing a seasonal outbreak every year. Occasionally, it comes as a disease pandemic with unprecedentedly high hospitalization and mortality. Currently, influenza is managed by vaccination and antiviral drugs such as Amantadine, Rimantadine, Oseltamivir, Peramivir, Zanamivir, and Baloxavir. However, the adverse side effects of these drugs, the rapid and unlimited variabilities of influenza viruses, and the emerging resistance of new virus strains to the currently used vaccines and drugs have necessitated the need to obtain more effective anti-influenza agents. In this review, essential oils are discussed in terms of their chemistry, ethnomedicinal values against flu-related illnesses, biological potential as anti-influenza agents, and mechanisms of action. In addition, the structure-activity relationships of lead anti-influenza EO compounds are also examined. This is all to identify leading agents that can be optimized as drug candidates for the management of influenza. Eucalyptol, germacrone, caryophyllene derivatives, eugenol, terpin-4-ol, bisabolene derivatives, and camphecene are among the promising EO compounds identified, based on their reported anti-influenza activities and plausible molecular actions, while nanotechnology may be a new strategy to achieve the efficient delivery of these therapeutically active EOs to the active virus site.

## 1. Introduction

Natural products (NPs) are chemical substances in the form of primary and secondary metabolites which are produced by living organisms such as plants, animals, marine organisms, bacteria, and fungi [1,2]. They are mostly referred to as small molecules or secondary metabolites, representing a unique scaffold of compounds that are distributed in a broad variety of chemical classes, such as alkaloids, cardiac glycosides, flavonoids, phenolics, saponins, sterols, and terpenoids [3]. Essential oil compounds form part of these classes as volatile and non-volatile aromatic substances, comprised mostly of terpenoids and some phenylpropanoid derivatives [4,5].

Chemically, terpenoids form the largest group of essential oils, such as monoterpenes (C10) and sesquiterpenes (C15) and a few diterpenes and phenylpropanoids [6]. Many of them have functional groups such as aldehyde, ether, ester, alcohol, carboxylic acid, phenol, amines, and amides [6].

Essential oils are naturally distributed in many higher plants and are most abundant in aromatic plant families, including Lamiaceae (Peppermint family), Myrtaceae (Eucalyptus family), Rutaceae (Citrus family), and Zingiberaceae (Ginger family) [7]. They have been reported in the seeds, fruits, peels, flowers, buds, leaves, young stems, barks, resins, woods, bulbs, roots, and rhizomes of many plants [8] and are extracted by methods of hydro-distillation, steam distillation, hydro-diffusion, solvent extraction, maceration, cold-press extraction, supercritical fluid (CO_2_) extraction, sub-critical liquid extraction, microwave-assisted extraction, and enfleurage [4,9].

The use of EOs predates modern history. They are one of the most important NPs that have ever been utilized by many cultures, for many centuries, around the world for domestic (cosmetics, perfumery, food, and beverages) and medicinal purposes [10]. Essential oils are popularly known to help relieve the airways during the cold (winter) or flu season, and evidence has emerged of their medicinal importance in the amelioration of respiratory tract infectious diseases, such as the common cold, pneumonia, and influenza [11]. The flu (influenza) is a disease that often occurs during the winter or cold season, and so can be referred to as seasonal influenza [12].

Influenza is an infectious zoonotic respiratory disease caused by the influenza A, B, C, and D viruses, where A and B are the common virus type that causes seasonal flu [13,14]. Influenza is characterized by the sudden onset of fever, dry cough, sore throat, runny nose, headache, and joint pains [12]. It is a major public health problem, accounting for about 3–5 million cases worldwide, with about a 10% annual death rate [12]. It is responsible for high hospitalization and deaths among high-risk individuals, such as the elderly and those with co-morbidities [15].

Vaccination remains a primary strategy to prevent and control influenza, due to the waning immunity that is associated with it [16]. So far, an effort made to reduce the burden of influenza through the administration of vaccines has yielded promising results, with more expectations for success. Vaccine selectivity is still a major issue because the currently available influenza vaccines are virus-type- and sub-type-specific [17]. In addition, many people are still in doubt about the harmlessness of vaccines due to cultural beliefs, myths, and conspiracy theories, thus causing vaccine hesitancy [18]. Therefore, there is a need for deliberate and continuous vaccine education and advocacy to help bring influenza to a halt.

Currently, four FDA-approved antiviral drugs of three different classes are recommended for use against the recently circulating influenza viruses, Baloxavir marboxil, Oseltamivir, Peramivir, and Zanamivir [19]. Despite the availability of these drugs, influenza is still on the rampage, taking its toll on human health and well-being. In fact, the problem is further exacerbated because the available drugs still lack strong antiviral activity against all the influenza virus strains; in addition, there is always an emergence of new resistant strains and the resurgence of old virulent strains that were not adequately reduced in past influenza outbreaks [20].

Therefore, there is an urgent need to identify lead antiviral agents that can be developed into new anti-influenza drugs. This review explores essential oils because of their age-long ethnomedicinal use across many cultures for the management of flu and other airway diseases. The chemical compositions of EOs are discussed with respect to their anti-influenza potentials. The mechanism of action and structure-activity relationship of the lead antiviral compounds are also highlighted in this study to identify the leading EO compounds that can be optimized as anti-influenza drug candidates.

## 2. Methodology

The study was carried out by an extensive, open (no duration set) literature review of articles in the BioMedCentral (BMC), Elsevier, Google Scholar, Hindawi, PubMed, Nature, ScienceDirect, Scopus, and Springer Nature databases, amongst others. The keywords for the search included “essential oils”, “essential oil compounds”, “essential oils and flu”, “influenza viruses”, “anti-influenza essential oils”, “anti-influenza essential oil compounds”, “mechanism of action of anti-influenza essential oil compounds”, and “structure-activity relationship of anti-influenza essential oil compounds”. Some inclusive criteria such as “natural volatile compounds”, “terpenes and terpenoids”, “aromatic terpenes”, “phenylpropanoid derivatives”, “antiviral essential oils”, “anti-flu essential oils”, and “anti-influenza terpenes”, were also used in the literature search. Compounds other than EO compounds, and diseases other than influenza or flu were excluded from the literature search. All the chemical structures were drawn with a Chem Draw Ultra^®^ 7.0 Software application, licensed by CambridgeSoft Corporation (Cambridge, MA 02140, USA), while diagrammatic representations were prepared with Microsoft PowerPoint Version 365 licensed by ©Microsoft Corporation (Johannesburg, South Africa).

To the best of our knowledge, this article gives a comprehensive and up-to-date account of the anti-influenza potentials of essential oils and their compounds and will add to the repository of anti-influenza essential oil compounds.

## 3. Essential Oils as an Integral Part of Natural Products

### 3.1. Natural Products

Natural products (NPs) are chemical substances that originate from plants, animals, marine organisms, fungi, and bacteria [21]. They are mostly secondary metabolites produced by living organisms and used by them for defense and adaptation purposes [22]. NPs represent a large group of diverse chemical entities with a broad spectrum of biological activities that have found many applications, notably in humans and veterinary medicines, food and agriculture, and cosmetics [22,23].

Natural products are a diversified group of natural substances, mostly secondary metabolites, discovered to provide a variety of health benefits in humans [24]. Evidence has emerged of the various biological and/or pharmacological activities of NPs [25,26]. Thus, they are regarded as drug leads because many known drugs are inspired by or developed from them [27]. For instance, the anti-influenza drugs, Favipiravir, Oseltamivir, Peramivir, and Zanamivir are nature-inspired [28].

Plants produce an enormous variety of NPs with diverse chemical structures [29]. These chemical entities fall into seven major categories: alkaloids, carbohydrates, specialized amino acids and peptides, flavonoids, polyketides and fatty acids, terpenoids and steroids, and phenylpropanoids [30] (Figure 1). A group of volatile and non-volatile, low molecular weight aromatic compounds, termed essential oils, can be found in the last three categories [4,5].

### 3.2. Plant-Derived Essential Oils

The International Organization for Standardization (ISO) defined plant-derived essential oils as products obtained from aromatic plants through specialized extraction methods without any significant change to their chemical compositions [31]. They are volatile and aromatic chemical substances, which are a mixture of fragrant and odorless compounds, mostly confined in the plant cytoplasm in the form of tiny droplets between cells and named after the plants from which they are derived [31,32].

Plant-derived EOs are produced in the glandular trichomes and other specialized secretory organs of plants, which include flowers, fruits, seeds, leaves, bark, and roots in the forms of bulbs and rhizomes [33]. They are usually plants’ secondary metabolites and are known to perform major ecological and physiological functions in plants, such as defense against herbivores and microbial pathogens (irritants and repellents), reduction in abiotic stress, allelopathy, inter-plant signaling, defense against pathogenic microorganisms (antimicrobial agents), and attraction of plant pollinators and seed dispersers for the purpose of plant reproduction and survival [34,35].

Well over 3000 EOs have been identified from about 2000 plant species belonging to aromatic families such as Asteraceae, Lamiaceae, Myrtaceae, Poaceae, Rosaceae, Rutaceae, Umbelliferae, and Zingiberaceae, amongst others [36], as presented in Table 1.

### 3.3. Biosynthetic Routes of Essential Oil Compounds

Essential oils are complex mixtures of hydrocarbons and their oxygenated and nitrogenous derivatives, derived from two different isoprenoid pathways in the secretory cells of plants [33], the acetate-mevalonate pathway for the formation of isoprenoids from isopentenyl pyrophosphate and dimethylallyl pyrophosphate in the cytoplasm, and the pyruvate pathway for the synthesis of 2-C-methyl-D-erythritol-4-phosphate in the plastids [76,77,78]. Thus, both pathways are mediated by different enzymatic reactions [76], as presented in Figure 2.

### 3.4. Classes of Essential Oils

Essential oils are comprised mostly of a mixture of low molecular weight, lipophilic, and sparingly polar organic compounds, and it is these properties that contribute to their high level of volatility [5]. Based on their chemical properties, they are structurally categorized as isoprenoids (terpenoids), a major EO group, while other minor groups include the phenylpropanoid derivatives [79].

Structurally, an isoprene unit (2-methylbutan-1,3-diene) is the building block upon which all isoprenoids are formed, according to the isoprene rule propounded by Breitmaier [80]. The isoprene unit (hemiterpene, C5 unit) through head-to-tail and tail-to-tail condensations, forms seven classes of terpenoids, monoterpenes (C10 isoprenoids), sesquiterpenes (C15 isoprenoids), diterpenes (C20 isoprenoids), sesterterpenes (C25 isoprenoids), triterpenes (C30 isoprenoids), tetraterpenes (C40 isoprenoids), and polyterpenes (at least C45 isoprenoids) [75,79,80]. However, many of the low molecular weight terpenoids (C5, C10, and C15 isoprenoids) and a few others (C20 and C25 isoprenoids) are naturally found to be EO compounds [77,81,82,83]. Essential oil compounds can be grouped into acyclic and cyclic isoprenoids, while the cyclic group is further sub-grouped into monocyclic, bicyclic, tricyclic, etc., based on the number of cyclic rings [5,84,85,86].

### 3.5. Medicinal Applications of Plant-Derived Essential Oils

For centuries, essential oils have continued to find applications in many cultures around the world for different purposes, such as foods, beverages, perfumes and cosmetics, and medicines [10]. Aromatherapy is an ancient but popular traditional medicine practice that uses EOs as the major therapeutic agent to treat many diseases [87]. Among the early accounts for the medicinal application of EOs was the practice of aromatherapy by the ancient Egyptians in 4500 BC, traditional Chinese medicine (TCM) in 3000 BC, and the ancient Indian Ayurvedic medicine around 2000 BC [10]. Later, ancient Greek scientists provided the first documented evidence for the medicinal application of EOs, such as cumin, peppermint, saffron, and thyme, around 450 BC [7]. An infusion of EO from the 16th-century rosemary plant (*Salvia rosemarinus*) later became a form of medicine after it was used by Queen Isabell to cure rheumatism in the court of King Louis XIV [88].

Essential oils have continued to find usefulness in disease prevention and management to date. Evidence has emerged concerning the medicinal application of aromatic herbs, their EO compounds, and other bioactive plant secondary metabolites for the management of common respiratory diseases (RDs), such as asthma, chronic obstructive pulmonary disease (COPD), pneumonia, and influenza, to mention a few [89]. The last few decades witnessed the development of herbal pharmacopoeias and monographs, some of which have validated the ethnomedicinal claims of aromatic herbs and their EOs as remedies for chronic RDs such as flu (influenza) and its associated ailments [90,91]. Therefore, there is a need for a better understanding of influenza and the biological potential of natural EOs against this malady.

## 4. Influenza (Flu)

Influenza is a contagious viral disease that affects the upper and lower respiratory tract [92]. Influenza viruses can be found in humans and some animals such as Aves and cattle, and are generally categorized as type A, B, C, and D influenza viruses [93,94]. The common types are the influenza A and B viruses, which affect humans and are mostly characterized as the seasonal flu [92]. Influenza A viruses are largely implicated in the flu pandemic and are a common cause of zoonotic infections, often characterized by virulent infections in humans [14,95]. The influenza C viruses are predominantly responsible for mild illness in animals and are rarely implicated in human epidemics [14,95]. Influenza D viruses mostly affect animals, with rare cases of human-to-human transmission [93]. Symptoms associated with influenza virus infections include a fever, sore throat, runny nose, cough, fatigue, and headache, owing to disease of the upper respiratory tract, while the lower respiratory tract may present with severe or acute pneumonia [96].

The influenza disease caused by the type A influenza virus of zoonotic origin, is a major public health concern, as it is responsible for both the common seasonal influenza epidemic (seasonal flu) and the sporadic and unpredictable (10–50 years of occurrence) global influenza pandemic outbreaks [97]. Seasonal influenza outbreaks typically occur during the winter season in temperate regions (Europe, Southern Africa), due to favorable conditions of low humidity and low temperatures [97,98]. However, in tropical countries, it is characterized by a complex pattern of occurrences due to an interplay of climatic factors such as temperature levels, hours of sunshine, and the level of rainfall [98].

On the other hand, pandemic influenza is characterized by a fast spread of the influenza A virus from the virus origin to the rest of the world in several waves over a short period, as witnessed in the first influenza pandemic of 1918 by the influenza A H1N1 virus strain, and subsequent pandemics of 1957, 1968, and 2009, caused by the Influenza A H2N2, H3N2, and H1N1 virus strains, respectively [97,99].

According to the World Health Organization, up to 1 billion influenza virus infection cases are reported annually, with about 4 million of the cases leading to severe illness, and around 400,000 reported deaths [95]. The most vulnerable groups are often infants between the ages of 0–9 months and adults 65-years-old and above [100].

Vaccination remains an effective means to reduce the burden of influenza [101]. The National Advisory Committee on Immunization (NACI) recommended the use of two classes of influenza vaccines, the Inactivated Influenza Vaccines (IIVs) and the Live Attenuated Influenza Vaccines (LAIVs) [102]. Just prior to the COVID-19 pandemic, it was reported that vaccination prevented an estimated 3.7 million cases of influenza, 105,000 influenza-related hospitalizations, and 6300 influenza-associated deaths worldwide [103]. However, more success in flu vaccination is still desired. Some key issues to be addressed include the complexity involved in predicting the pattern of seasonal influenza, reduced vaccine efficacy based on repeated annual immunization, an antigenic mismatch between the developed vaccines and the circulating virus strains, an age difference of the different cohorts involved in vaccination, and the issue of variability in the virulence level of different seasonal strains of the virus [104,105,106].

Even though vaccination is the most effective means of reducing the burden of influenza, antiviral drugs can be very useful in delaying the spread of new pandemic viruses, and they have also been found useful for the treatment of critically ill influenza patients [107]. There have been significant strides in the development of influenza antiviral drugs (IADs), and there are currently three classes of FDA-approved IADs: M2 proton channel antagonists, neuraminidase inhibitors, and polymerase acidic endonuclease inhibitors [108]. The drugs Amantadine and Rimantadine, are M2 proton channel antagonists, which used to be effective for the treatment of influenza A virus infection but have lost their efficacies over the years due to the emergence of more virulent strains of the type A virus, such as the 2009 H1N1 influenza A virus [19]. The FDA-approved neuraminidase inhibitors such as Oseltamivir, Peramivir, and Zanamivir are more efficacious and less toxic for the management of influenza than the M2 proton channel antagonists [109]. However, these drugs are associated with adverse effects, such as skin rash, diarrhea, anaphylactic reaction, headache, nausea, vomiting, cough, and gastritis [108]. Baloxavir is a cap-dependent, polymerase acidic endonuclease inhibitor which is similar in potency to neuraminidase inhibitors, except for the fact that it is newer and offers a different mechanism of action from the earlier developed neuraminidase inhibitory drugs [108,110].

Based on the rapid and unlimited variabilities of influenza viruses and the emerging resistance of new influenza virus strains to the currently used drugs, there is a dire need to discover more lead anti-influenza agents with a novel mechanism of action and develop (synthesize and optimize) more effective analogs from the already existing ones [109]. Natural products, including EOs, continue to offer an inexhaustible reservoir of bioactive compounds as lead therapeutic agents for the management of diseases. Some EOs and their compounds have been reported to demonstrate remarkable biological activities against a wide range of viruses, including influenza viruses [111]. It is against this backdrop that the anti-influenza potentials of EOs and their compounds were discussed, vis a viz the mechanism of action and structure-activity relationships of lead antiviral compounds, to source newer anti-influenza agents.

## 5. Essential Oils as Potential Anti-Influenza Agents

### 5.1. Anti-Influenza Properties of Plant-Derived Essential Oils and Their Compounds

Evidence has emerged on the anti-influenza potentials of many aromatic plants that are used for the treatment of flu and flu symptoms (cold, cough, sore throat, bronchitis, and pneumonia) by various ethnomedicines [112].

For instance, the EOs of *Cynanchum stauntonii* roots demonstrated an in vitro activity against Influenza A/NWS/33 (H1N1) virus at an IC_50_ value of 64 µg/mL and selectivity index of 8, with the main EOs used comprising (E,E)-2,4-decadienal, 3-ethyl-4-methypentanol, 5-pentyl-3H-furan-2-one, (E,Z)-2,4-decadienal, 2(3H)-furanone,dihydro-5-pentyl, and caryophyllene oxide [113]. Further investigation revealed considerable inhibitory effects on influenza-induced deaths with 40, 70, and 100% survival rates when administered 50, 150, and 300 mg/kg doses of the EO, respectively, in male albino mice [113].

The leaf EOs of *Melaleuca alternifolia* (tea tree oil) contain terpinen-4-ol, terpinolene, and α-terpineol, which showed considerable in vitro activity against the influenza A virus in MDCK cells by interference with acidification of intra-lysosomal compartment [114]. *Mosla dianthera* is an aromatic herb used in the TCM to treat colds, coughs, nasal congestion, bronchitis, fever, and headache [115]. The EOs derived from the aerial part of *M. dianthera* exhibited significant in vivo inhibitory activity against the influenza A virus at 90–360 mg/kg body weight in mice, with elemicin, thymol, β-caryophyllene, iso-elemicin, asarone, and α-caryophyllene implicated as the major active ingredients [112].

The in vitro antiviral activities of *Citrus bergamia* and *Eucalyptus globulus* EOs against the influenza A H1N1 virus have also been reported, in which *E. globulus* vapor EOs reduced viral infection by 78%, with no cytotoxicity, while that of *C. bergamia* reduced the viral cytopathic effect with no cytotoxicity [116]. The major EOs characterized in the former are limonene, linalyl acetate, and linalool, while the latter contained 1,8-cineole, γ-terpinene, *p*-cymene, and α-thujene [116]. In addition, thujone- and α-pinene rich Cedar (*Thuja plicata*) EO vapor demonstrated over 90% inhibitory activities against some membrane-containing influenza A H1N1 and H3N2 and B viruses within 10 min of exposure to the viruses in vitro [117]. Other anti-influenza aromatic plants and their major EO components are presented in Table 2, while the structures of some anti-influenza EO compounds are shown in Figure 3.

### 5.2. Mechanisms of Action and Structure-Activity Relationships of Some Lead Anti-Influenza Essential Oil Compounds

Basically, the molecular mechanisms of action of lead anti-influenza agents can be summed up under two major categories: those agents that target influenza virus proteins or genes and those that target the various components within the hosts for replication and propagation [107]. These mechanisms can be used to further categorize anti-influenza agents (virus inhibitors). First, entry and attachment (fusion) inhibitors, which are commonly used as an adjuvant in the preparation of anti-influenza vaccines [132,133]. The aerial EO of *Melaleuca alternifolia*, known as tea tree oil (TTO), has been shown in an in silico simulation study to interfere with the entry and fusion activities of the influenza virus [116]. The anti-influenza activity has been attributed to its hydroxylated monoterpenes, terpinen-4-ol, and α-terpineol [114]. Other known groups are hemagglutinin inhibitors [134], M2 ion channel protein inhibitors [135], endosomal and lysosomal inhibitors, also implicated in the TTO [125], protease inhibitors [136], RNA polymerase inhibitors [137], pathway inhibitors [138], neuraminidase inhibitors [139], non-structural protein inhibitors [107], caspase inhibitors [140], glycoprotein/glycosylation inhibitors [141], phospholipase inhibitors [142], release inhibitors [143,144], and autophagy [145]. Natural antiviral agents including EO compounds can act as inhibitors during the influenza virus activity stages of binding, penetration, uncoating, genome replication, assembly, and release of the virus (Figure 4); thus, they may offer considerable protection and efficacy as anti-influenza agents [146].

Three major compounds, curdione, curcumol, and germacrone, were implicated in the antiviral EO components of the TCM Zedoary oil [147]. The compounds impaired influenza A (H1N1) virus replication in vitro and in vivo, with germacrone exhibiting the highest anti-H1N1 effect [147]. Germacrone was shown to activate the transcription of interferon genes and protect the peripheral cells from influenza virus infections [147]. It also showed a marked decrease in the expression of antiviral proteins, RIG-I, IFNs, OAS, IRF3/7, MX, and EIF2AK2/PKR, viral replication, and viral load, with increased TAP1 expression, inhibited TAK1 phosphorylation, and consequently inhibited the NF-κB signaling and intrinsic antiviral responses (Figure 5) [147,148]. The biological properties of germacrone have been linked to its ketone and non-conjugated double bonds (Figure 6) [149].

Some anti-influenza active bisabolane-type sesquiterpenoids from turmeric oil (*Curcuma longa*) have also been reported [150]. Generally, turmeric oil is used in ethnomedicine for the treatment of flu-related and/or airway inflammatory diseases, such as bronchitis, pneumonia, and influenza [151]. The compounds significantly acted as pathway inhibitors against the influenza A/PR/8/34 (H1N1) virus replication in MDCK and A549 cells in vitro [150]. The compounds act by inhibiting the expression of pro-inflammatory cytokines (IL-6, IL-8, IP-10, and TNF-α), and regulating the activity of the NF-κB/MAPK and RIG-1/STAT-1/2 signaling pathways [150]. The presence of ketone, α, β-unsaturation, and presence of an electron-withdrawing group (OH, OCH_3_, NH_2_, SH, and halogens) have been reported to influence the bioactivity of this group of compounds (Figure 6) [152].

Eucalyptol (1,8-cineole), a monoterpenoid principally from Eucalyptus plants, is another lead anti-influenza agent to discuss [153]. Eucalyptus oil is used in traditional medicines as a remedy for respiratory ailments [154]. Eucalyptol is a major EO component of the oil, and it has been shown to exert considerable protection against influenza viral infection in vivo [155]. The oil efficiently decreased the levels of cytokines, IL-4, IL-5, IL-10, and MCP-1 in nasal lavage fluids, as well as the levels of IL-1β, IL-6, and necrosis factors TNF-α and IFN-γ in the lung tissues of mice infected with the influenza A virus [153]. It also reduced the expression of the inflammatory response, NF-kB, p65, intercellular adhesion molecule (ICAM)-1, and vascular cell adhesion molecule (VCAM)-1 in lung tissues [154]. The findings thus suggest that eucalyptol is capable of augmenting protection against influenza virus infection in mice by inhibiting pulmonary inflammatory responses in the tissues [154].

In another study, eucalyptol (12.5 mg/kg) demonstrated lower viral titers, less pulmonary edema, less weight loss, less inflammation, a lower mortality rate, and a longer survival time when it was co-administered with 0.2 µg of haemagglutinin influenza vaccine, compared to when the vaccine was administered alone [156]. The mechanism of action of eucalyptol has been reported to be an increase in the antiviral activity of IRF3 as well as the IκBα- and JNK-dependent inhibitory effect of IRF3 on the NF-κB p65 and NF-κB proinflammatory signaling pathways [156,157]. The presence of an epoxy functional group and the unique inter-atomic distance between the R1-C-O-C-R2 of eucalyptol have been linked to its remarkable biological effects (Figure 6) [158].

In a recent study, isocaryophyllene acetamides (ICAs) and some S-containing derivatives of caryophyllene oxide (caryophyllane thiosesquiterpenoid, CTS) were shown to inhibit the replication of rimantadine-resistant influenza virus A/California/07/09 (H1N1) pdm09 and influenza virus A/Puerto Rico/8/34 (H1N1) strains, respectively [159,160]. Due to the natural bicyclic framework of ICAs (Figure 6, they are known to show marked anti-influenza activity by blocking the M2 protein of the influenza virus and by inhibiting the cleavage of hemagglutinin [160]. This led to an aggregation of the virus and lysosomal vacuole membranes and virus inactivation [160]. Gyrdymova and others demonstrated the influence of the S-containing functional group on anti-influenza activity, showing that bisulfide-containing CTS compounds possess high virus-inhibitory activities and suggesting S-containing derivatives of caryophyllene oxide as promising substrates for the design of newer anti-influenza and/or antiviral agents [161].

Some caryophyllene derivatives (Ginsamides, GAs) were reported to demonstrate dose-dependent virus inhibition and subtype-specific virus-inhibiting activity (IC_50_ = 0.15 µM) against the influenza virus H1N1 and H1N1pdm09 strains in a pool of influenza virus A/Puerto Rico/8/34 (H1N1) strains in MDCK in vitro cell cultures [162]. Ginsamides showed considerable in vivo protective ability against the virus at 150 mg/kg/day and inhibited the fusogenic activity that is typical of influenza A/Puerto Rico/8/34 (H1N1) viruses [163]. According to the report, GAs can act as lead inhibitors against the viral infection of normal cells and may offer the host an opportunity to maintain a complete immune response [162,164]. Structurally, the bicyclic backbone and the amide functional group of GAs (Figure 6) are known to confer a high level of antiviral and anti-influenza activities [107,161,163].

Eugenol and citronellol are major EO compounds of *Cinnamomum zeylanicum* and *Pelargonium graveolense,* respectively [165,166]. The combined EOs demonstrated in vitro antiviral activity (MIC = 100 3.1 µL/mL) against the influenza A (H1N1) virus [167]. It acted by targeting the virus surface before and during the adsorption event in the viral lifecycle, thus making it a natural neuraminidase inhibitor [167]. Structurally, eugenol and citronellol contain phenolic hydroxyl and primary alcohol functional groups, respectively (Figure 6), which confer some biological properties, such as antioxidant, anti-inflammatory, and antiviral activities, amongst others [168,169].

A novel camphor-based anti-influenza agent, camphecene, has been reported to cause a significant decrease in the number of influenza virions fusing their envelopes with endosomal membranes [170]. This nitrogen-containing camphor derivative has been reported to possess unique chemical properties that bind it effectively to the active sites of hemagglutinin (HA), acting as an HA inhibitor, and thus causing a decrease in viral pathogenicity [170]. Based on the pharmacokinetic study, camphecene demonstrated a remarkable decrease in virus titer in the lungs and mortality at 7.5 mg/kg, following a 6 h dose regime in vivo [171]. It also demonstrated an additive effect with Tamiflu, a synthetic anti-influenza drug, which suggests it is an anti-influenza drug candidate [170,171]. Several analogs of this compound have been synthesized, and the structure-activity relationship analysis suggests that camphecene analogs should bear an oxygen atom with a short linker (C2–C4), either as a hydroxyl or ketone group, or as part of a heterocycle (Figure 6), for optimal anti-influenza activity [172].

## 6. Future Perspectives and Conclusions

As it has been shown in this review, there have been deliberate efforts by scientists to exploit the EOs of individual aromatic plants or groups of plants for their anti-influenza potentials, partly due to the emergence of more antigenic influenza viruses, the increased lethality of influenza disease outbreaks, the reduced effectiveness level of vaccines and drugs, and the ethnomedicinal consideration of natural products for alternative medicines. At best, Choi reported 62 plant EOs for in vitro antiviral activity against three selected influenza virus strains [128]. Vimalanathan and Hudson evaluated the in vitro anti-influenza activities of EO vapors obtained from nine aromatic plants [167], while a recent report by Wani and colleagues showed antiviral activities against the influenza A (H5N1) virus by EOs derived from bergamot, cinnamon, lavender, lemongrass, thyme, and citrus [119]. Therefore, a discussion of the various associated aromatic plants and their chemical products became necessary. The common botanical sources of EOs, their chemical classification and biogenetic routes, and the antiviral properties and molecular mechanism of action of some EO compounds were major items of discussion in our review to identify drug candidates that can be optimized to mitigate the ferocity of antigenically distant and vaccine-/drug resistant strains of the influenza viruses. Thus, this may be one of the very few reviews that extensively discusses sourcing anti-influenza agents specifically from EOs and their aromatic compounds.

It is our opinion that a multi-level approach should be put in place to resolve the serious health crisis caused by seasonal and pandemic flu outbreaks. First, there should be region-specific influenza vaccination programs in influenza virus-originating areas [98,173]. For instance, annual vaccination campaigns should be initiated about 5 months apart in Northern and Southern China, and influenza surveillance should be significantly improved in the mid-latitude provinces due to the complexity associated with seasonal patterns in these regions [98]. In addition, there is an urgent need to develop universal influenza vaccines that can offer protection against antigenically distant influenza viruses [174].

A novel antiviral approach, termed small interfering RNA (siRNA) vector technology, can be adopted to bring about a multiple-fold reduction in viral titer shed [175]. However, this method has only been validated in an in vitro assessment. The medicinal application of EOs and their lead compounds as anti-influenza agents or, simply put, for therapeutic use, has generated much interest in recent times. Currently, the U.S. FDA indicates EOs for use as cosmetic and food supplements or drugs [176]. Therefore, there is a need to validate the herbal raw materials, including ascertaining the reputation of their sources and standardizing both the extraction process and the final products (EOs), for quality assurance purposes [176,177,178].

Some EO compounds, including 1,8-cineole, eugenol, germacrone, thiol- and amide derivatives of caryophyllene oxide, curcumol, terpinen-4-ol, and bisabolane-type sesquiterpenoids, have all shown considerable potential as influenza drug candidates in this study. However, there are more comprehensive mechanistic studies as well as detailed clinical evaluations on these lead EO compounds. For instance, the exact effects of germacrone on the influenza virus life cycle need to be critically evaluated to provide a proof-of-concept for the development of novel influenza virus inhibitors [107].

In addition, there are conflicts regarding the role that individual compounds play in the overall EO antiviral activity. For instance, *Eucalyptus globulus* and *Salvia officinalis* both contain 1,8-cineole (eucalyptol) as the major component. However, the former plant oil was reported to have a strong activity (IC_50_ < 3.1 μg/mL) against the influenza (H1N1) virus, while the latter was poorly active [167]. Therefore, the various compounds making up an EO should be evaluated for their individual anti-influenza properties in both in vitro and in vivo settings so that the probable biological role of each compound can be determined.

A milestone achievement worthy of mention is the use of newer formulation strategies, such as nanobiotechnology, to offer a site-specific and target-oriented delivery approach to treating diseases [179]. This novel technology has also been adopted in recent times for the efficient delivery of therapeutically active EOs [180,181] and uses encapsulation strategies to develop lipid-based delivery systems, such as solid nanoparticles, nanostructured lipid carriers, liposomes, and micro- and nano-emulsions [176]. The nano formulation techniques reduce volatility and increase bioavailability while improving chemical stability and reducing toxicity, thus overcoming the limitations of high volatility, hydrophobicity, instability, and the risk of toxicity associated with the pharmaceutical application of EOs [176].

Despite the beneficial attributes offered by nanobiotechnology, the half-lives of nano-formulated bioactive EOs need to be improved while their pharmacokinetic parameters need to be optimized [181]. This can be achieved by searching for other chemical derivatives with a prolonged period of anti-influenza activity and by optimizing the application schedule, as was achieved in the synthesis of novel anti-influenza camphecene analogs from camphor [171,172].

In conclusion, essential oils are an integral part of natural products with medicinal potential for the management of illnesses such as influenza (flu) and other respiratory diseases. There is an urgent need to exploit nature for more novel anti-influenza agents, vis a viz conducting preclinical and clinical evaluations on established antiviral EO compounds, for the development of newer influenza drugs. This will require collaborative research efforts for health solutions so that good health and well-being can be attained in real-time.

## Figures and Tables

**Figure 1 molecules-27-07797-f001:**
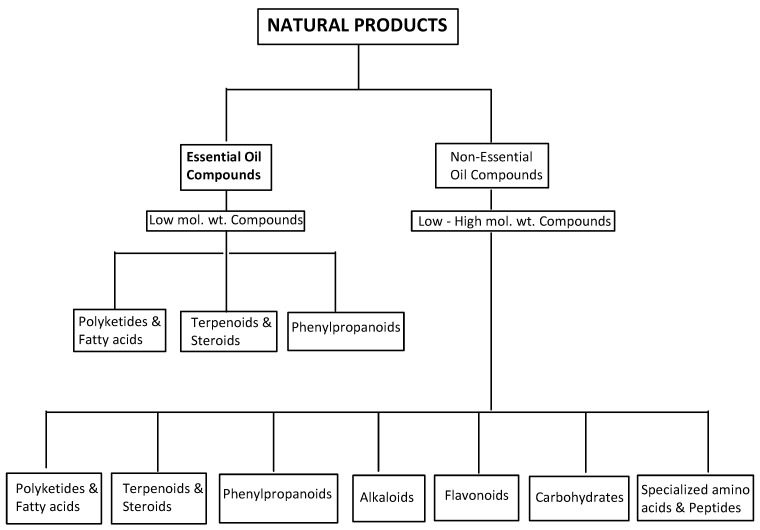
An overview of natural products, showing the essential oil compounds.

**Figure 2 molecules-27-07797-f002:**
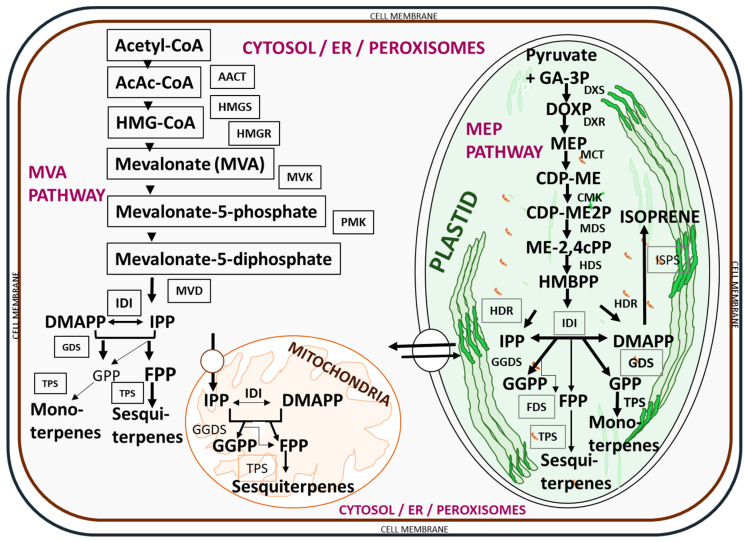
Biosynthetic pathways for essential oil compounds in plants. A list of enzymes involved in the biosynthesis includes acetoacetyl-CoA (AcAc-CoA), acetoacetyl-CoA thiolase (AACT), CDP-ME kinase (CMK), DOXP reducto-isomerase (DXR), DOXP synthase (DXS), farnesyl diphosphate synthase (FDS), geranyl diphosphate synthase (GDS), geranyl geranyl diphosphate synthase (GGDS), (E)-4-hydroxy-3-methylbut-2-enyl diphosphate reductase (HDR), (E)-4-hydroxy-3-methylbut-2-enyl diphosphate synthase (HDS), 3-hydroxy-3-methylglutaryl-CoA (HMG-CoA), HMG-CoA reductase (HMGR), HMG-CoA synthase (HMGS), isopentenyl diphosphate isomerase (IDI), isoprene synthase (ISPS), 2-C-methyl-D-erythritol-4-phosphate cytidylyltransferase (MCT), 2-C-methyl-D-erythritol 2,4-cyclodiphosphate synthase (MDS), mevalonate diphosphate decarboxylase (MVD), mevalonate kinase (MVK), phosphomevalonate kinase (PMK), and terpene synthase (TPS). The key intermediates (compounds) involved in the biosynthetic process include: 4-(cytidine 50-diphospho)-2-C-methyl-D-erythritol (CDP-ME), 4-(cytidine 50-diphospho)-2-C-methyl-D-erythritol phosphate (CDP-ME2P), isopentenyl diphosphate (IPP), dimethylallyl diphosphate (DMAPP), farnesyl diphosphate (FPP), geranyl geranyl diphosphate (GGPP), geranyl diphosphate (GPP), glyceraldehyde-3-phosphate (GA-3P), 2-C-methyl-D-erythritol 2,4-cyclodiphosphate (ME-2,4Cpp), 2-C-methyl-D-erythritol-4-phosphate (MEP), (E)-4-hydroxy-3-methylbut-2-enyl diphosphate (HMBPP), and 1-deoxy-D-xylulose 5-phosphate (DOXP). The pathways were adapted from Nagegowda [76] and redrawn with copyright permission by the Federation of European Biochemical Societies © 2010 Elsevier B.V. Publication.

**Figure 3 molecules-27-07797-f003:**
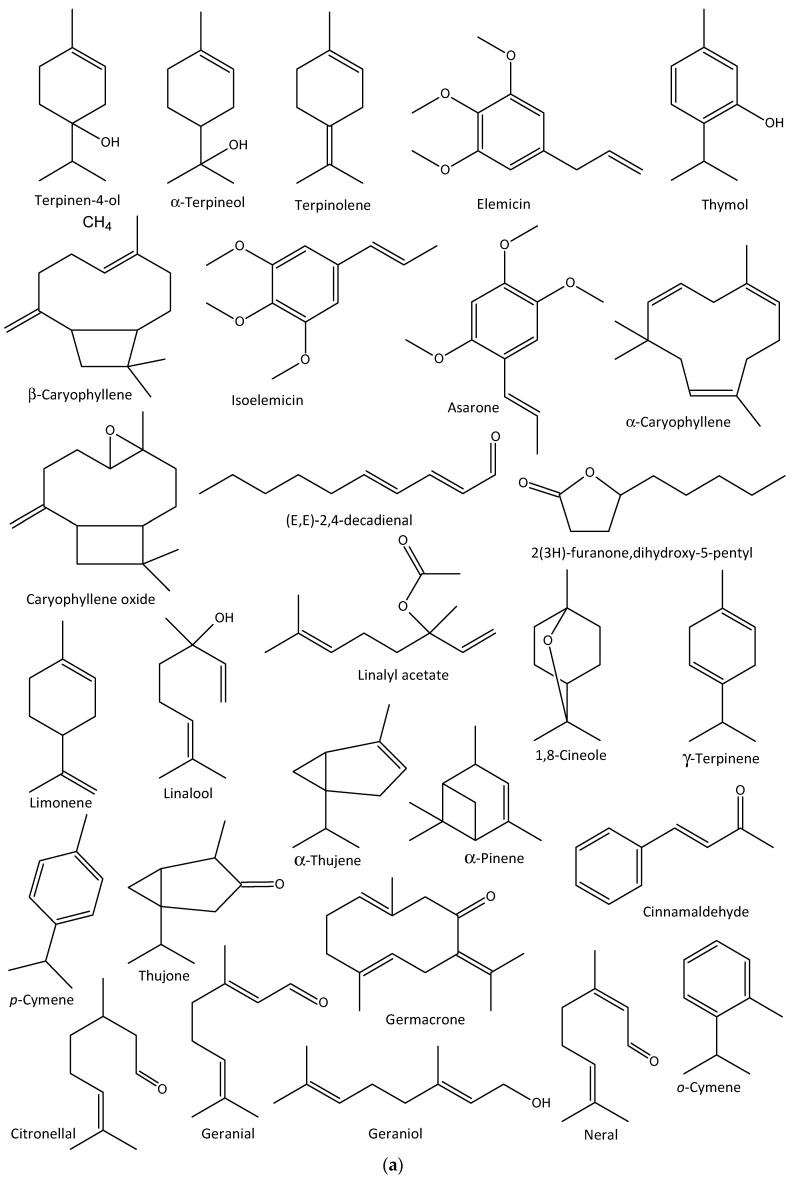
Structures of some bioactive essential oil compounds (**a**,**b**) that have been implicated against some influenza viruses.

**Figure 4 molecules-27-07797-f004:**
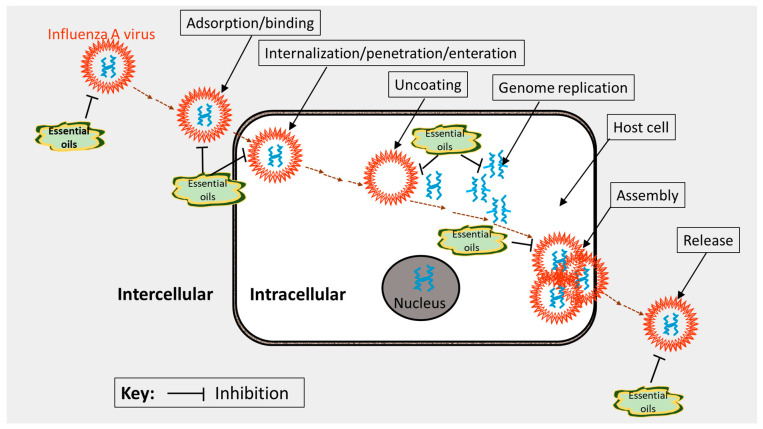
Plausible target sites (mechanisms of action) of essential oils during the influenza virus lifecycle. The illustration was adapted from Ma and Yao [146] and redrawn with copyright permission by ©MDPI, Basel, 2020.

**Figure 5 molecules-27-07797-f005:**
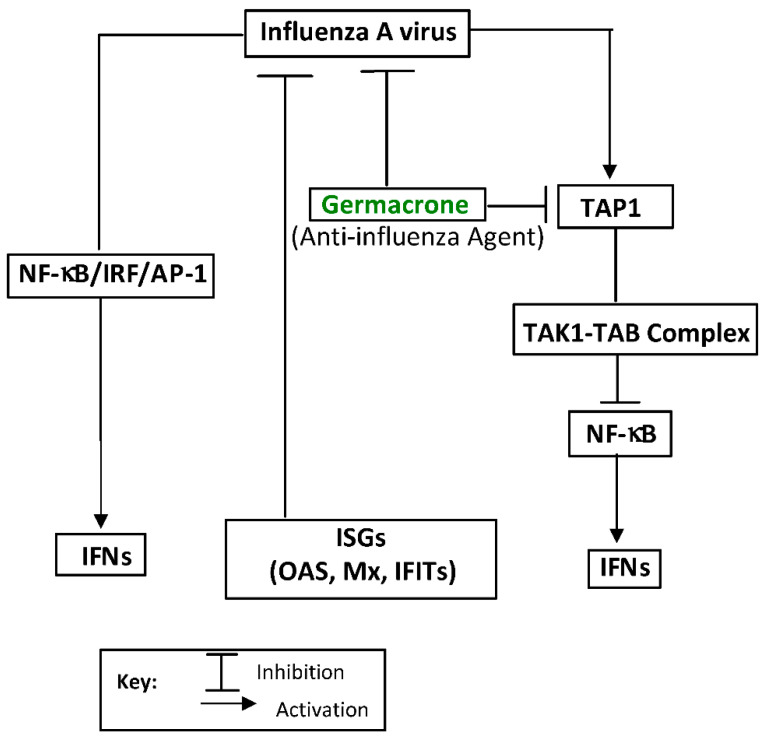
Inhibition of TAP1 expression (pathway inhibition) by germacrone, a lead antiviral essential oil compound against the influenza A (H1N1) virus, and its replication: a mechanism of action. Nuclear factor kappa-light-chain-enhancer of activated B cells (NF-κB), interferons (IFNs), and interferon regulatory factors (IRFs) regulate many aspects of innate and adaptive immune responses, including driving anti-viral responses. Interferon stimulating genes (ISGs) are critical effectors of IFN response to viral infection. Transcription factor (AP-1) regulates the inflammatory gene expression in response to viral infections. The transporter associated with antigen processing 1 (TAP1), transforming growth factor-β-activated kinase 1 (TAK1), and TAK1 binding protein (TAB) are also involved. The illustration was adapted from Li et al. [147] and redrawn with copyright permission by © Elsevier BV, The Netherlands, 2020.

**Figure 6 molecules-27-07797-f006:**
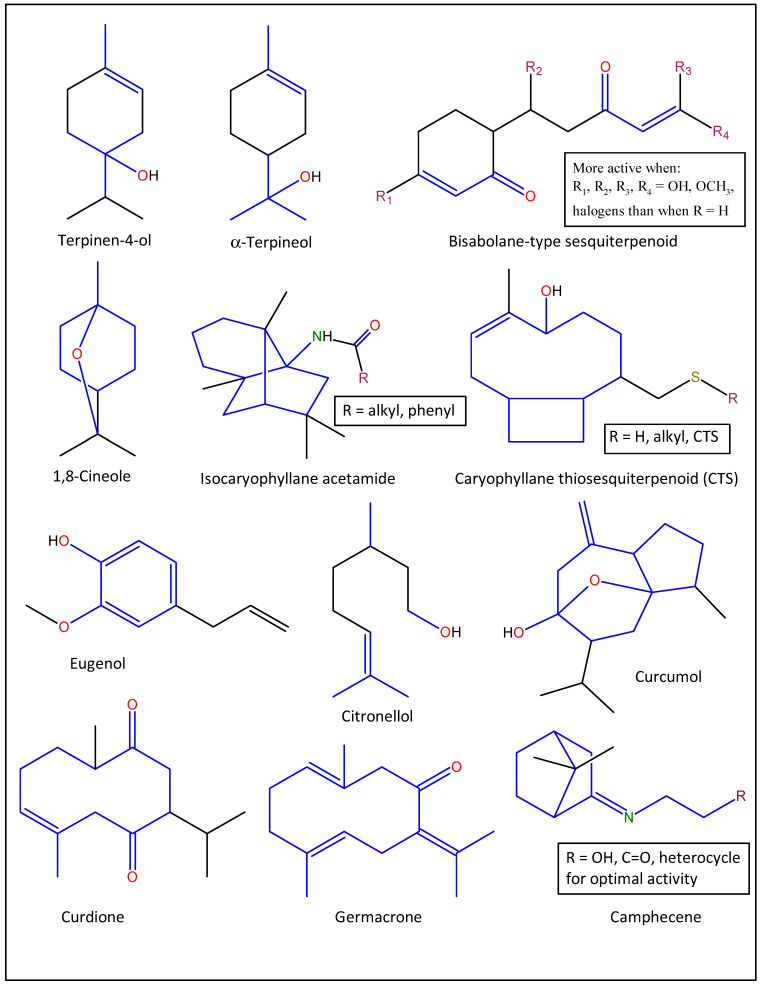
Structures of some lead anti-influenza essential oil compounds showing their active moieties (colored).

**Table 1 molecules-27-07797-t001:** Some Essential Oil-Bearing Plant Families with their Major Essential Oil Compounds.

Plant Family	Species (Part Used)	Major Essential Oil Constituents	Extraction Method	Reference
Apiaceae (Umbelliferae or Carrot family)	*Angelica archangelica* (Flowers)	α-Pinene, β-phellandrene, limonene, and ρ-cymene	Hydro-distillation (HD) for 2 h	[37]
	*Daucus carrota* (Seeds)	Sabinene, carotol, (Z)-β-farnesene, elemecin, and β-bisabolene	HD for 2 h	[38]
Asteraceae (Sunflower family)	*Matricaria chamomilla* (Flowers)	α- and β-Farnesene, α-bisabolol, chamazulene, germacrene D, and spiroether	HD for 4 h	[39]
	*Achillea millefolium* (Leaves, flowers, and seeds)	Eucalyptol, camphor, α-terpineol, β-pinene, sabinene, 1,8-cineole, artemisia ketone, linalool, α- and β-thujone, camphor, borneol, bornyl acetate, (*E*)-β-caryophyllene, germacrene D, caryophyllene oxide, β-bisabolol, δ-cadinol, and chamazulene	HD for 10 h (industrial) HD for 3 h (small scale)	[40,41]
Brassicaceae (Mustard family)	Aethionema sancakense (Aerial parts)	α-Humulene, camphene, and heptanal	HD for 3 h	[42,43]
	Brassica oleraceae (Aerial parts)	Dimethyl disulfide, dimethyl trisulphide, allyl isothiocyanate, dimethyl tetrasulfide, and 1-hexanol	HD for 12 h (industrial)	
Burseraceae (Torchwood family)	*Boswellia sacra* (Oleogum resin)	(*E*)-β-Ocimene, 1-β-pinene, 2-β-pinene, camphene, sabinene, α-thujene, limonene, myrcene, α-pinene, 2-carene, (*Z*)-β-ocimene, δ- and γ-cadinene and caryophyllene oxide, β-elemene, and α-copaene.	Steam distillation (SD) in water/ethylene glycol (1:9) for 2 h	[44]
	*Commiphora myrrha* (Oleogum resin)	Furanoeudesma-1,3-diene, lindestrene, curzerene, β-elemene, germacrene B, and germacrone	Supercritical fluid extraction (SFE) with CO_2_ and vacuum extraction with dichloromethane by Ultrasonic bath	[45,46]
Cupressaceae (Conifer or Cypress family)	*Cupressus sempervirens* (Aerial part)	α-Pinene, δ-3-carene, limonene, and α-terpinolene	HD for 3 h	[47]
	*Juniperus communis* (Berries)	α- and β-Pinene, myrcene, sabinene, and limonene	Commercial SD	[48]
	*Juniperus virginiana* (Heartwood)	(-)-α- and (+)-β-Cedrene, (-)-thujopsene, and (+)-cedrol	Commercial SD	[49]
Lamiaceae (Labiatae or Mint family)	*Salvia leucophylla* (Leaves and flowers)	1,8-Cineole, camphor, camphene, and α- and β-pinene	HD for 2 h	[50]
	*Ocimum basilicum* (Aerial part)	Methyl cinnamate, linalool, β-elemene, and camphor	HD for 3 h	[51]
	*Mentha piperita* (Aerial part)	Menthol and menthone, (±)-menthyl acetate, 1,8-cineole, limonene, β-pinene, and β-caryophyllene	HD for 3 h	[52]
	*Mentha spicata* (Aerial part)	Carvone, limonene, 1,8-cineole, β-pinene, cis-dihydrocarvone, and dihydrocarveol	HD for 3 h	[53]
	*Rosmarinus officinalis* (Aerial part)	ρ-Cymene, linalool, γ-terpinene, thymol, α- and β-pinene, and eucalyptol	HD for 3 h	[54]
	*Thymus vulgaris* (Aerial part)	Thymol, ρ-cymene, γ-terpinene, and caryophyllene oxide	HD for 3 h	[55]
Lauraceae (Laurel family)	*Cinnamomum camphora* (Stem bark, leaves, and fruits)	D-Camphor, 1,8-cineole, α-terpineol, linalool, safrole, γ-terpinen, isoterpinolene, 1,3,8-ρ-menthatriene, terpinen-4-ol, α-terpineol, eugenol, β-cadinene, and α-cubebene	HD for 6 h	[56]
	*Cinnamomum zeylanicum* (Leaves, stem bark, fruits, and, roots)	(*E*)-Cinnamaldehyde, linalool, β-caryophyllene, eucalyptol, eugenol, carvacrol, ρ-cymene, α-humulene, δ-cadinene, and α-pinene	HD for 3 h	[57]
	*Laurus nobilis* (Leaves)	1,8-Cineole, sabinene, linalool, α-terpinyl acetate, α-pinene, α-terpineol, methyl-eugenol, neoiso-isopulegol, eugenol, β-pinene, and γ-terpinene	HD for 3 h	[58]
Myrtaceae (Myrtle family)	*Eucalyptus* species (Leaves)	1,8-Cineol, α-pinene, spathulenol, trans-pinocarveol, ρ-cymene, globulol, cryptone, β-phellandrene, viridiflorol, borneol, limonene, and isospathulenol	HD for 4 h HD for 3 h	[59,60]
	*Myrtus communis* (Berries)	Geranyl acetate, 1,8-cineole, α-terpinyl acetate, methyleugenol, linalool, α-terpineol, β-caryophyllene, α-humulene, trans-caryophyllene oxide, and humulene epoxide II	HD for 4 h	[61]
	*Melaleuca alternifolia* (Leaves and terminal branch)	Terpinen-4-ol, γ-terpinene, 1,8-cineole, α-terpinene, α-terpineol, ρ-cymene, and α-pinene	Commercial SD	[62]
Pinaceae (Pine family)	*Picea mariana* (Twigs/bark and needles)	α- and β-Pinene, β-phellandrene, 3-carene, limonene, α-terpineol, trans-pinocarveol, terpinen-4-ol, verbenone, borneol, and pinocarvone	SD and HD for 6 h each	[63]
	*Pinus eldarica* and *P. peuce* (Twigs/bark, needles, and pollens)	D-Germacrene, α- and β-pinene, trans-(*E*)-caryophyllene, γ-terpinene, limonene, caryophyllene oxide, drimenol, β-myrcene, camphene, bornyl acetate, and δ-cardinene	HD for 4 h each	[64,65]
Poaceae (Grass family)	*Cymbopogon citratus* (Leaves)	Geranial, neral, β-myrcene, geranyl acetate, isopulegol, and citral (comprising cis-isomer geranial and trans-isomer neral)	SD for 3 h	[66]
	*Cymbopogon martini* (Leaves)	Linalool, α-terpineol, geranyl isobutyrate, geraniol, myrcene, β-caryophyllene, geranyl acetate, (E,Z) farnesol, and geranyl hexanoate	SD for 3 h	[67]
	*Cymbopogon nardus* (Leaves)	6-octenal, citronellal, geranial, geraniol, citronellol, and neral	HD for 3 h	[68]
Rutaceae (Citrus family)	*Citrus limon* (Leaves and fruit peels)	β-Pinene, limonene, linalool, α-terpineol, linalyl acetate, acetate geranyl, nerolidol, acetate neryl, and farnesol	SD for 3 h	[69]
	*Citrus paradisi* (Leaves and fruit peels)	D-Limonene, β-myrcene, γ-terpinene, β-phellandrene, furanoid, caryophyllene, and cis-limonene oxide	HD for 4 h	[70]
	*Citrus sinensis* (Leaves, and fruit peels)	D-Limonene, β-pinene, 3-carene, β-phellandrene, and linalool	Cold-press molecular distillation	[71]
Zingiberaceae (Ginger family)	*Curcuma longa* (Rhizomes)	Ar-turmerone, α-turmerone, curlone, and ar-curcumene	Commercial SD	[72]
	*Elettaria cardamomum* (Leaves and rhizomes)	β-Caryophyllene, γ-terpinene, α- and β-pinene, myrcene, p-cymene, limonene, 1,8-cineole, linalool, 4-terpineol, and α-terpinyl acetate	HD for 3 h	[73]
	*Zingiber officinale* (Rhizomes)	Citral (geranial and neral), α-zingiberene, camphene, 1,8-cineol, α-farnesene, β-sesquiphellandrene, α-terpinene, α-terpineol, 4-terpineol, gingerols, zingerone, paradol, gingerdiones, gingerdiols, shogaols, and zingerines	HD for 5–6 h	[74,75]

**Table 2 molecules-27-07797-t002:** Some plant-derived essential oil compounds and their anti-influenza properties.

Aromatic Plant (Part)	Major Essential Oil Component	Anti-influenza Activity	Reference
*Cinnamomum verum* (Cortex)	trans-Cinnamaldehyde	Inhibited the growth of influenza A/PR/8 virus in vitro at 200 µM. Inhalation (50 mg/cage/day) and nasal inoculation (250 microg/mouse/day) of EOs significantly increased survival rates over 8 days to 100% and 70%, respectively, in vivo.	[118]
*Citrus reshni* (Unripe fruit peels)	Limonene (82.4%) and linalool (7.2%)	In vitro antiviral activity (IC_50_ = 2.5 µg/mL) against Avian influenza virus A (H5N1) strain.	[119,120]
*Curcuma aeruginosa* (Rhizomes)	Germacrone	In vitro antiviral activity (EC_50_ = 6.0 µM) against influenza A/H1N1/H3N2 virus strains and the influenza B virus in a dose-dependent manner. Germacrone demonstrated effective protection of mice from lethal infection and reduced the virus titers in the lung at 100 mg/kg in mice. Its co-administration with oseltamivir showed an additive effect on the inhibition of influenza virus infection, both in vitro and in vivo.	[121]
*Eucalyptus globulus* (Stems and leaves)	1,8-Cineol (84.2%) and *o*-cymene (8.0%)	Intranasal co-administration of 1,8-cineole with the influenza vaccine provides cross-protection against influenza virus infection in vivo.	[122,123]
*Fortunella margarita* (Fruits)	α-Terpineol (55.5%), carvone (5.7%), *t*-carveol (5.5%), muurolene (5.5%), and citronellal (5.01%)	In vitro antiviral activity (IC_50_ = 6.8 µM) against Avian influenza A (H5N1) virus.	[124]
*Melaleuca alternifolia* (Aerial part)	terpinen-4-ol (36.7%) and γ-terpinene (22.2%), and α-terpinene (10.1%)	Antiviral activity (ID_50_ = 0.0006%v/v) against the influenza A⁄ PR⁄ 8 virus subtype H1N1 with activity attributed to terpinen-4-ol.	[125]
*Melissa officinalis* (Leaves)	Geranial (38.3%), neral (26.1%), geraniol (8.1%), caryophyllene oxide (5.5%), citronellal (4.5%), neryl acetate (3.9%), and geranyl acetate (3.3%)	Inhibited influenza virus replication through different replication cycle steps. Co-administration of EOs with oseltamivir showed a synergistic activity with the EO, especially when oseltamivir concentration was under 0.005 mg/ml.	[126,127]
*Pimpinella anisum* (Fruits)	trans-anethole (82.8%), estragole (8.2%), and linalool (2.7%)	In vitro activity against influenza A/WS/33 virus (IC_50_ < 100 µg/mL) infected MDCK cells, inhibited the formation of a visible cytopathic effect.	[128]
*Pogostemon cablin* (Leaves)	patchoulol, caryophyllene, pogostol, α-, β-, γ- and δ-patchoulene, seychellene, cycloseychellene, α- and β-bulnesene, α- and β-guaiene, and norpatchoulenol	In vitro activity (IC_50_ of 4.0 µM) against influenza virus A (H2N2). Significant increase in the survival rate and survival time within the 20–80 mg/kg doses for the 7-day post-infection period in mice.	[129,130]
*Salvia sclarea* (Flowers)	Linalyl acetate (61.2%), linalool (22.1%), α-Terpineol (4.2%), and geranyl acetate (2.4%)	Inhibitory activity (>52%, IC_50_ < 100 µg/mL) against influenza A/WS/33 virus-infected MDCK cells, inhibited the formation of a visible cytopathic effect.	[128]
*Thymus mastichina* (Leaves)	1,8-Cineole (64.6%), linalool (15.3%), β-pinene (5.8%), and α-pinene (4.2%)		
*Waldheimia glabra* (Whole plant)	α-Bisabolol (20.2%), valeranone (11.8%), chamazulene (9.9%), spathulenol (8.2%), β-caryophyllene (6.1%), and caryophyllene oxide (5.2%)	Cytopathic effect against influenza H3N3 virus (IC_50_ = 88.8 µg/mL) and cytotoxicity against MDCK cells (CC_50_ = 252 µg/mL).	[131]

## Data Availability

Not applicable.

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
