# Peer review of "Essential Oils and Their Compounds as Potential Anti-Influenza Agents"

_molecules, 2022, doi:10.3390/molecules27227797_

Round 1
Reviewer 1 Report
This review is devoted to the topic of essential oils, as well as their components as potential anti-influenza agents. The article is written in a clear and accessible language. The main ideas expressed by the authors are clear. The relevance of the study is beyond doubt, since in the last two years the topic of antiviral drugs (in various forms) has also become more important than before. This review well systematizes knowledge in this area. However, there are some points that need to be improved:
1. Technical:
1.1 The main text should be bold without highlighting. Only headings and subheadings are designated in this way.
1.2 Text alignment should be full width.
1.3 The font size of the text in tables should be equal to (or less than) the size of the main font in the text. This applies to all tables.
1.4 There are many spelling and punctuation errors in the text. I recommend that the authors carefully re-read the text on this subject.
2. Main content:
2.1 The abstract is written too modestly. Please expand on this.
2.2 Table 1. It is necessary to add a column to it indicating the solvent with which this essential oil was obtained.
2.3 In table 1, you can enter data from the work (10.3390/molecules27186129), as well as make a link to this work.
2.4 In the text of the article, you can consider other reviews on the topic of essential oils and indicate the relevance and uniqueness of the review made by the authors. What is the peculiarity and advantage of the authors' review over other reviews on this topic?
Author Response
Point 1.1: The main text should be bold without highlighting. Only headings and subheadings are designated in this way.
Response 1.1. The highlight has been removed in the main text except for headings and subheadings
Point 1.2. Text alignment should be full width
Response 1.2. Text alignment has been revised to show full width
Point 1.3. The font size of the text in tables should be equal to (or less than) the size of the main font in the text. This applies to all tables
Response 1.3. The tables have been re-formatted to font size 10, in line with the journal format
Point 1.4. There are many spelling and punctuation errors in the text. I recommend that the authors carefully re-read the text on this subject
Response 1.4. The manuscript has been proof-read and corrected for incorrect spellings, inconsistencies and punctuations.
Point 2.1 The abstract is written too modestly. Please expand on this.
Response 2.1. The abstract has been expanded
Point 2.2. Table 1. It is necessary to add a column to it indicating the solvent with which this essential oil was obtained
Response 2.2. A column on the extraction methods used has been added to Table 1
Point 2.3. In table 1, you can enter data from the work (10.3390/molecules27186129), as well as make a link to this work
Response 2.3. I have included the work by Demirpolat et al. 2022 (10.3390/molecules27186129) and Morales-López et al. 2016, to represent the aromatic plants in the family Brassicaceae.
Point 2.4. In the text of the article, you can consider other reviews on the topic of essential oils and indicate the relevance and uniqueness of the review made by the authors. What is the peculiarity and advantage of the authors' review over other reviews on this topic?
Response 2.4. Our review has been put in perspective of other similar/related studies, and the peculiarity and significance of our study have been emphasized. Please check lines 105-107 and lines 474-491.
Reviewer 2 Report
The manuscript must be rejected because of:
1. Extensive English editing is required.
2. fig 2 is very complex and contains a lot of details, please modify.
3. fig. 3 is basic knowledge, and does not relate to the main core of the review, please omit.
4. Please limit the drawn structures to the active constituents of the essential oil components related to the titled review.
5. Please strictly follow the author guidelines.
6. Most of the figures are copied from other sources, do authors have legal permission.
Author Response
RESPONSE TO REVIEWER
Point 1. Extensive English editing is required.
Response 1. The manuscript has been proof-read by an English Editor and also subjected to online grammar check. Please check through for highlights in yellow.
Point 2. fig 2 is very complex and contains a lot of details, please modify.
Response 2. The details in fig 2 have been modified by first mentioning the key enzymes involved in the biosynthetic pathways, followed by the major intermediates (compounds) involved in the biosynthetic process.
Point 3. fig. 3 is basic knowledge, and does not relate to the main core of the review, please omit.
Response 3. Figure 3 has been expunged from the manuscript.
Point 4. Please limit the drawn structures to the active constituents of the essential oil components related to the titled review.
Response 4. Figures 4-6 have been expunged from the manuscript, so that the focus is solely on the topic of discussion, i.e., the active constituents alone.
Point 5. Please strictly follow the author guidelines.
Response 5. The manuscript has been revised according to the author guidelines.
Point 6. Most of the figures are copied from other sources, do authors have legal permission.
Response 6. All figures included in the manuscript were not lifted or copied verbatim but redrawn and modified for use. This is why the authors and publishers of such thoughts were cited.
Thank you.
Reviewer 3 Report
This review covers a specific aspect of the link between essential oils and their value as biological agents against influenza. While there have been many reviews of biological activity of natural products, this specific approach is quite new, and therefore valuable. The sourcing of data is very wide, going beyond hard scientific peer-reviewed journal papers to include softer information from other sources and general experiences. This approach provides an excellent starting point for deeper studies in this area, something that is urged by the authors. I therefore recommend acceptance of the review for publication in molecules, essentially in present form. However, some gentle editing would help to eliminate some minor grammatical infelicities.
Author Response
RESPONSE TO REVIEWER'S COMMENT
Point/Suggestion. Some gentle editing would help to eliminate some minor grammatical infelicities.
Response. The manuscript has been edited for grammatical errors, punctuations, spelling errors and others. The manuscript has been revised accordingly.
Thank you.
Round 2
Reviewer 1 Report
Accepted
Reviewer 2 Report
The manuscript has been sufficiently improved